# Review and Evaluation of *Ostertagia ostertagi* Antibody ELISA for Application on Serum Samples in First Season Grazing Calves

**DOI:** 10.3390/ani13132226

**Published:** 2023-07-06

**Authors:** Johannes Charlier, Tong Wang, Sien H. Verschave, Johan Höglund, Edwin Claerebout

**Affiliations:** 1Kreavet, Hendrik Mertensstraat 17, 9150 Kruibeke, Belgium; twang@kreavet.com; 2Laboratory of Parasitology, Faculty of Veterinary Medicine, Ghent University, Saliburylaan 133, 9820 Merelbeke, Belgium; sien_verschave@fas.harvard.edu (S.H.V.); edwin.claerebout@ugent.be (E.C.); 3Department of Molecular and Cellular Biology, Harvard University, 52 Oxford Street, Cambridge, MA 02138, USA; 4Department of Biomedical Sciences and Veterinary Public Health, Swedish University of Agricultural Sciences, SE-750 07 Uppsala, Sweden; johan.hoglund@slu.se

**Keywords:** nematode, cattle, diagnosis, serum ELISA, *Ostertagia*

## Abstract

**Simple Summary:**

Gastrointestinal (GI) nematode infections are a significant health and welfare threat to calves in pasture-based rearing systems. Sustainable control requires an efficient and cost-effective diagnostic tool. The serum pepsinogen assay is a long-established tool to monitor the infection level. However, the relatively high cost and lack of standardisation hinder the broad implementation of this method. Here, the more cost-efficient and robust *Ostertagia ostertagi*-Ab ELISA is evaluated as a potential alternative diagnostic method in first-season grazing (FSG) calves. We performed both a literature review of studies where this method has been applied in FSG calves and conducted field studies in Belgium and Sweden to compare results from the *O. ostertagi*-Ab ELISA with the serum pepsinogen assay. We conclude that the *O. ostertagi*-Ab ELISA test is a valuable monitoring tool in FSG calves and could complement or replace the serum pepsinogen assay.

**Abstract:**

The *O. ostertagi*-Ab ELISA assay is widely used as a diagnostic tool for monitoring gastrointestinal (GI) nematodes using milk samples from adult dairy cows. This assay is potentially also useful to analyse serum samples from first-season grazing (FSG) calves, providing a more cost-effective and robust diagnostic technique than the current serum pepsinogen assay. However, a comprehensive evaluation of its use in serum samples from FSG calves has not yet been conducted. In this study, we first reviewed the available scientific literature in which the *O. ostertagi-*Ab ELISA was applied to serum samples from FSG calves. Then, a field study was conducted to compare results from the *O. ostertagi*-Ab ELISA assay with a serum pepsinogen assay on a set of 230 serum samples from 11 commercial dairy herds (seven in Belgium and four in Sweden). The literature review showed an increase in mean antibody levels, expressed as optical density ratio (ODR) values, from <0.4 (early grazing season) to values of 0.7–1.1 (late grazing season). Three out of five studies found a negative correlation between *O. ostertagi* antibody levels measured during the late grazing season and weight gain, while the other two studies found no correlation between the two variables. Our field studies showed a good correlation between *O. ostertagi* antibody levels and the results from the pepsinogen assay. Both indicators were negatively related to average daily weight gain in the Belgian herds, but not in the Swedish herds. Overall, the results suggest that the *O. ostertagi*-Ab ELISA test can be a useful tool in FSG calves and could replace the use of the serum pepsinogen assay at the end of the grazing season for general monitoring purposes.

## 1. Introduction

Serum pepsinogen concentration is considered one of the most reliable parameters for monitoring gastrointestinal (GI) nematode infections in cattle during their first grazing season. Previous studies have shown that herd mean serum pepsinogen concentration correlates with abomasal worm burden, pasture infectivity, the intensity of chemoprophylaxis, and weight gain (e.g., [1,2,3,4,5]). Despite the diagnostic value of serum pepsinogen levels, two problems have prevented widespread use and uptake as a routine diagnostic tool in practice: (i) the cost of the pepsinogen assay and (ii) a lack of reproducibility. The serum pepsinogen assay is an enzymatic assay, and although the method has been simplified over the years [6], the procedure remains more complex with longer incubation times and a higher cost than conventional ELISAs, which are more easily automated in commercial laboratories.

As an alternative to pepsinogen determination in cattle, a commercial assay for the quantification of antibodies against GI nematodes is available (Svanovir^®^
*O. ostertagi*-Ab, Indical Biosciences). The results of this test are expressed as optical density ratio (ODR) [7]. Antibody levels against GI nematodes correlate reasonably well with pepsinogen levels and have been proposed as an alternative for monitoring purposes [8,9]. In addition, the Svanovir^®^ assay is considered highly reproducible [7]. Therefore, the use of serum antibody measurement for monitoring GI nematode infections has the potential to overcome some of the limitations of the serum pepsinogen assay. However, the Svanovir^®^ assay was developed primarily for use on bulk tank milk [10], and a comprehensive evaluation for its use in first-season grazing (FSG) calves has not yet been conducted. Furthermore, pepsinogen levels provide a more direct assessment of the clinical impact of GI worm infection (abomasal damage) than antibodies, which reflect previous GI nematode larval exposure and do not necessarily correlate with current infection or morbidity.

The objectives of this study were: (i) to review the scientific literature in which the Svanovir^®^ *O. ostertagi-*Ab ELISA has been applied to serum samples from FSG calves and evaluate whether interpretive criteria can be proposed, and (ii) to compare the diagnostic value of *O.ostertagi*-AB ELISA and pepsinogen tests on a set of 230 serum samples. Based on the overall result gained from these two approaches, guidelines for the use and interpretation of the *O. ostertagi* ELISA on serum samples from FSG calves are proposed.

## 2. Methods

### 2.1. Literature Review of Svanovir^®^ O. ostertagi-Ab ELISA Used on Serum Samples

The Svanovir^®^ *O. ostertagi-*Ab ELISA has been on the market since 2007. Although it is only licensed for use on bulk tank milk samples, researchers have also used it on serum samples. We searched 2 databases (Pubmed and Google Scholar) to find publications in which the Svanovir^®^ *O. ostertagi-*AB ELISA was used on serum from FSG calves. The following keyword combinations were used (cattle OR bovine OR cow OR heifer OR calf OR calves) AND (nematode OR ostertagia OR Cooperia) AND (diagnos* OR ELISA OR pepsinogen OR svanov*). Only publications from 2007 onwards were considered. The search was conducted up to articles published on or before August 2022. The literature found was first subjected to a title-based selection, followed by a second selection based on reading the full text.

For each publication in which the Svanovir^®^ *O. ostertagi-Ab* ELISA was applied to bovine sera, the following information was noted: (i) whether the assay was performed on sera from adult or growing animals; (ii) production type of the animals: beef vs. dairy; (iii) range and statistical distribution of the ELISA results; and (iv) correlations observed with indicators of parasite infection and/or animal productivity.

### 2.2. Comparative Analysis of O. ostertagi Antibody and Pepsinogen Levels Measured in Sera of Dairy Calves

Data were collected from FSG calves from seven commercial dairy herds in Belgium (2011) and four commercial dairy herds in Sweden (2022). Faecal and serum samples were collected from the calves at the beginning, middle, and end of the grazing season. Serum samples were subjected to the Svanovir^®^ *O. ostertagi-*Ab ELISA according to the manufacturer’s instructions and to the serum pepsinogen assay according to the micro-method of Dorny and Vercruysse [6]. For the ELISA, sera were diluted 1:140, and results were expressed as ODR [11]. Serum pepsinogen assay results were expressed in Units of Tyrosine (U Tyr). Faecal egg counts (FEC) were determined using a modified McMaster technique with a sensitivity of 10 eggs per gram (EPG). In addition, animals were weighed with a calibrated scale at each visit, and anthelmintic treatments during the grazing season were recorded. Further details on the Belgian herds and the associated data have been described by Rose et al. [12].

The different parasitological parameters (i.e., serum pepsinogen concentration, serum *O. ostertagi* antibody level, and FEC) at the individual level were compared by comparative plotting and Spearman rank correlation. Since serum pepsinogen levels are usually used for diagnosis at herd level, the mean value of the parasitological parameters per herd-sample point was calculated. These values are referred to as the “herd mean” in the following. A ROC analysis was performed to determine the threshold in the herd mean *O. ostertagi* ELISA result that best predicts a mean serum pepsinogen concentration of more than 2 U Tyr. As there was only one observation where the herd mean serum pepsinogen concentration was higher than 2.5 U Tyr, no other thresholds could be tested.

We created three generalized linear mixed models (GLMMs) with average daily weight gain (ADWG) as the response variable and each of the three parasitological parameters (late-season ODR, late-season pepsinogen, and mid-season FEC), as a fixed effect, respectively. In all three models, location (Belgium and Sweden) was treated as a second fixed effect and herd (farm) as a random effect. For the third model that used FEC as a fixed effect, negative binomial GLMM was used as it gave a better model diagnostic result. As serum pepsinogen levels can drop rapidly after anthelmintic treatment, all observations made after anthelmintic treatment were removed from the analysis for statistical comparisons in Belgium. This was not taken into account in the Swedish data, as the treatments were carried out in the first part of the grazing season, and reinfection was possible for at least 3 months before the end of the grazing season. All analyses were carried out with the statistical software package R (version 3.2.2 R Core Team 2015, Vienna, Austria).

## 3. Results

### 3.1. Literature Review

The search strings yielded 810 publications in PubMed. A title-based selection yielded 35 potentially relevant publications. A second full-text-based selection of 15 publications reporting results on the use of the Svanovir^®^ *O. ostertagi-Ab* ELISA on bovine sera. Of these, 10 publications applied the assay to sera coming from calves, while the remaining 5 applied it to sera from adult cattle only—the latter were disregarded, and only the 10 publications with data on calves were considered further. The search in Google Scholar with the described search terms resulted in >40,000 hits. Therefore, a new search was performed with the specific search term “ostertagia” AND “svanova” AND “serum.” This yielded 72 publications. The same title and abstract-based selection as before were applied to select the most relevant publications, and this resulted in the retrieval of a relevant PhD dissertation [13], making a total of 11 selected studies.

The 11 relevant studies are summarised in Table 1. Briefly, in all studies where animals were sampled several times during the grazing season, an increase in mean ODR values was observed of values mostly less than 0.4 to values ranging from 0.7 to 1.1 at the end of the grazing season. However, in one study, ODR values were already relatively high (about 1.2) at the beginning of the grazing season and continued to rise to about 1.6 near the end of the season [13]. In young beef calves, there was evidence of an effect of passive transfer of maternal antibodies on calf ODR levels [14]. In experimentally challenged calves, ODR values differed distinctively between groups of calves that were either (i) trickle-challenged by infective larvae, (ii) trickle-challenged and treated with an anthelmintic, or (iii) uninfected [15]. In a large-scale epidemiological study in western Canada, ODR levels correlated with meteorological variables [16]. Four studies examined the correlation between parasitological indicators and weight gain. Two studies found negative correlations of weight gain with FEC and pepsinogen concentration, but no correlation with *O. ostertagi* antibody levels [5,13]. In contrast, two French studies [17,18] divided animal groups into low (<0.7 ODR) and high (≥0.7 ODR) exposure groups based on the mean antibody levels measured. In the high exposure groups, average daily weight gain (ADWG) was negatively correlated with increasing *O. ostertagi* antibody levels, and there was a significant improvement in ADGW in treated animals compared to untreated animals. On the other hand, no correlation between *O. ostertagi* antibody levels and ADWG, and no effect of treatment on ADWG was observed in the low-exposure groups.

### 3.2. Evaluation of the Svanovir^®^ O. ostertagi-Ab ELISA on Sera from Dairy Calves

#### 3.2.1. Data Description

The variables studied are summarised in Table 2. The frequency distribution of these variables at the second sampling point is shown in Appendix A, illustrating high variation in parasitological parameters and weight gain both within and between farms. FEC remained mostly low (<200 EPG) and was mostly highly skewed. Variation between farms was particularly high for weight. This was largely explained by the different ages of the animals at turnout on pasture. On the farms with a high body weight, FECs remained very low, even when the other parasitological parameters indicated moderate to high infection. The average ODR of the Belgian and Swedish herds was 0.66 and 0.49, while the average pepsinogen level was 1.7 and 1.5 U Tyr, respectively.

#### 3.2.2. Clinical Observations

Among the Belgian farms, no clinical signs of parasitic disease were observed throughout the study on Farms 1–3 and 7. Coughing was observed on Farms 4 and 6. The diagnosis of lungworm was confirmed by coprological analysis (Baermann technique/method) and urged anthelmintic treatment of the whole group in September and August, respectively. On Farm 5, poor growth, dull hair coat, and diarrhoea were observed, whereupon the animals were treated with an anthelmintic in September. No clinical signs were observed in the Swedish herds during this study.

#### 3.2.3. Correlation with Parasitological Parameters

The course of the mean serum pepsinogen concentration and the *O. ostertagi* antibody level during the grazing season is shown in Figure 1. Overall, there was a good visual correlation between both variables. At the last sampling point in Belgian Farms 2, 5, and 6, *O. ostertagi* antibody levels continued to increase, while pepsinogen levels decreased. However, in these cases, the animals had only been recently stabled/housed (Farm 2) or had recently received anthelmintic treatment (Farm 5 and 6). Finally, on Farm 7, the average serum pepsinogen level increased to >3 U Tyr, while the *O. ostertagi* antibody level decreased slightly.

The Spearman rank correlation (R) between the *O. ostertagi* antibody level, the serum pepsinogen concentration, and FEC on the animal and the mean values (by sample date) over the entire grazing season and at the end of the grazing season is shown in Table 3.

ROC analysis revealed that a threshold of 0.72 ODR in the mean *O. ostertagi* antibody level was best suited to detect a mean serum pepsinogen concentration > 2 U Tyr. At this threshold, sensitivity and specificity were 84% and 78%, respectively.

#### 3.2.4. Correlation with Weight Gain

The relationship between late-season parasitological parameters and weight gain is shown in Figure 2. Overall, both late-season antibody levels (*p* = 0.002, slope ± standard error = −0.537 ± 0.173) and late-season pepsinogen levels (*p* < 0.001, slope ± standard error = −0.227 ± 0.064) were negatively associated with ADWG. However, a significant interaction between the two parasitological parameters and location indicated that this negative relationship was present only in the Belgian herds and not in the Swedish herds (*p* = 0.007 and *p* = 0.004 for antibody and pepsinogen levels, respectively). Overall, an increase of 0.1 in the late-season ODR was associated with a weight loss of 54 g/day. Similarly, an increase of 0.1 U Tyr resulted in a weight loss of 23 g/day. No significant association was found between mid-season FEC and ADWG (*p* = 0.997).

## 4. Discussion 

The systematic review showed that the Svanovir^®^ *O. ostertagi* Ab-ELISA has already been used in a considerable number of studies on calf sera. The review found that *O. ostertagi* antibody levels typically increase gradually over the grazing season, but that there is significant variation between geographical regions. Based on these studies, an ODR threshold of 0.5 could be suggested as an indicator of GI nematode exposure. However, in a Scottish study, the detected antibody levels were already high before the grazing period. A possible explanation is the presence of passively transferred maternal antibodies in the serum of the calves, as suggested by Höglund et al. [14].

Our field study confirms the correlation between mean herd serum pepsinogen concentration and mean *O. ostertagi* antibody level in FSG cattle. The mean *O. ostertagi* antibody level started at an ODR value < 0.5 and increased to 0.6–1.2 by the end of the grazing season. In an attempt to find a practical indicator for GI nematode infections with an economic impact, we identified what ODR corresponds to a high serum pepsinogen concentration in our data. In general, a mean serum pepsinogen concentration above 2.5 or 3.0 U Tyr is considered an indicator of subclinical production losses caused by GI nematodes [4]. However, this threshold was only exceeded in one of the herds and could, therefore, not be further evaluated. In our study, a mean ODR of 0.72 discriminated between mean serum pepsinogen concentrations higher and lower than 2 U Tyr with reasonable sensitivity and specificity. Further studies with data from highly infected herds are needed to investigate an economic threshold for the Svanovir^®^ *O. ostertagi*-Ab ELISA on serum in FSG calves. Such data are not readily available as the economic threshold for the serum pepsinogen assay was only exceeded in 2–6% of herds in Sweden, Germany, and Belgium [23]. It is interesting to note that despite the generally low serum pepsinogen levels recorded, these were negatively correlated with weight gain, suggesting that nematode-related weight losses begin before the accepted economic threshold in the pepsinogen assay is reached.

Our study also highlights the differences between serum pepsinogen concentration and serum *O. ostertagi* antibody level as a diagnostic parameter, making the method of choice dependent on the situation or the purpose. Both parameters correlate negatively with ADWG when they are measured late season, but pepsinogen concentration shows a rapid decrease after anthelmintic treatment or housing of the animals. Serum pepsinogen might therefore be a better parameter for monitoring GI nematode infections during the grazing season to decide and evaluate an anthelmintic treatment. However, it is poorly suited to assess GI nematode infection once animals are housed, unless samples are taken shortly (<14 days) after housing. In contrast, *O. ostertagi* antibody levels do not drop suddenly after anthelmintic treatment or housing and are therefore better suited to assess past GI nematode exposure if samples are taken during the housing period. The negative correlation between ADWG and pepsinogen levels reported here was confirmed in another study [5], with the difference that they used mid-season pepsinogen levels. Conversely, another study did not find a correlation between ADWG and either antibody or pepsinogen levels [13]. However, the author did not specify when the animals were sampled. According to two other studies, *O. ostertagi* antibody levels seem to be a valuable parameter to assess the impact of GI nematode exposure or treatment on ADWG, but only if the ODR exceeds a certain level (≥0.7) before treatment [17,18]. Interestingly, this threshold (0.7 ODR) is similar to the threshold (0.72 ODR) found in our study to discriminate herds with a mean serum pepsinogen concentration > or <2 U Tyr. Also, Ploeger et al. [1,2] and Charlier et al. [11] found negative correlations between *O. ostertagi* antibody titres and weight gain in FSG or carcass weight in adult suckler cows, respectively. Although our study overall showed a significant influence of late-season antibody and pepsinogen levels on daily weight gain, there is a significant interaction between the location (Belgium/Sweden) and antibody or pepsinogen levels. Based on the Swedish data alone, no association was found between late-season antibody or pepsinogen levels and weight gain, which is consistent with the fact that only a subset of the studies in the literature review found such a negative relationship. The consistency and influencing factors of this relationship between parasitological parameters and production indices in FSG calves could be the subject of further investigation, but the effects of year-specific weather patterns leading to the possibility that the negative relationship only occurs in certain time windows, as well as nutritional, genetic and other differences in husbandry have already been mentioned as reasons for the variability [2,24].

## 5. Conclusions

The *O. ostertagi* antibody ELISA test has been increasingly used in studies over the past 15 years to monitor GI nematode infection in FSG calves. The group mean results correlate well with other parameters of GI nematode exposure in FSG calves (serum pepsinogen concentration and FECs). Test results also often correlate with weight gain, but not consistently. The test can complement and, in some cases, replace the serum pepsinogen assay for monitoring GI nematode infections, especially to assess past GI nematode exposure during the housing period.

## Figures and Tables

**Figure 1 animals-13-02226-f001:**
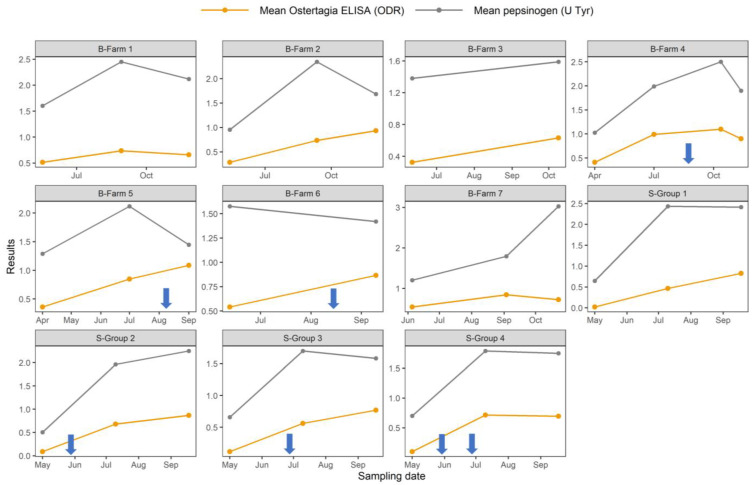
The course of the serum pepsinogen concentration and *O. ostertagi* antibody level (ODR) in cattle during their first grazing season on seven Flemish and four Swedish dairy farm herds. Blue arrows indicate the date of anthelmintic treatment of the group.

**Figure 2 animals-13-02226-f002:**
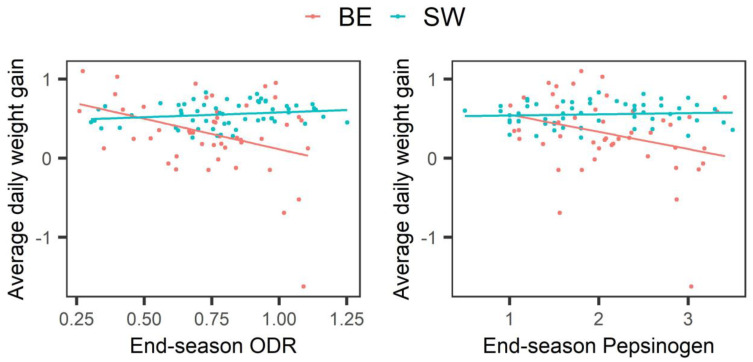
Relationship between two parasitological parameters and average daily weight gain (kg). Only observations that were taken before anthelmintic treatment were included. The red and blue lines in each plot show the line of best fit for Belgium (BE) and Swedish (SW) data, respectively.

**Table 1 animals-13-02226-t001:** Studies applying the Svanovir^®^ *O. ostertagi*-Ab ELISA on the serum of calves with a summary of the main findings.

Reference	Matrix (Dilution)	Age (Months)	Production Type	Number of Animals (Number of Herds/Auction Markets/Experiments)	Region/Country	Main Findings
Forbes et al., 2009, [15]. Vet Parasitol 162, 295–305	Plasma (1/140)	4–5	dairy	25 (1)	UK	Mean ODR values of animals that received a trickle challenge (dose rate 10,000 *O. ostertagi* larvae/day) from D0 to D55 with eprinomectin treatment at D56 reached maximum levels (0.7) at D56 and were significantly higher than in animals that received the same challenge but received eprinomectin treatment at D0 and D56 (mean max. ODR = 0.4). The mean ODR of uninfected controls remained close to 0 during the study.
Höglund et al., 2009, [5]. Vet Parasitol 164, 80–88	Serum (1/50 or 1/100)	N.A.	beef	330 (8)	Sweden	Serum samples were collected at intervals of 3 or 4 weeks throughout the grazing season. Mean ODR values ranged from 0.2 to 0.4 at the beginning of the grazing season and reached maximum levels between 0.7 and 1.0 from 12 weeks post turnout, and remained high for the remainder of the grazing season. Significant negative correlation between individual FEC and serum pepsinogen concentration and mid-season weight gain, but no correlation between weight gain and individual ODR.
A. Jackson, 2013, [13]	Plasma (1/140)	5–14	beef	90 (6)	Scotland	Mean ODR values start at ca. 1.2 ODR at the beginning of the grazing season and rise up to ca. 1.6 ODR at the end of the grazing season. Negative correlation between individual pepsinogen or FEC and weight gain, but no correlation between weight gain and individual ODR.
Höglund et al., 2013, [14]. Vet Rec 172, 472	Serum (1/50 or 1/100)	N.A.	beef	27 (2)	Sweden	Mean ODR gradually increased over the grazing season from 0.1 to 1.0 for early-born (December-February) calves. For late-born calves (February-April), it initially decreased from 0.51 and then increased again to 0.81. Importance of passive transfer of antibodies in late-born calves.
Colwell et al., 2014 [19]	Serum (1/140)	8	beef	328 (1)	West Canada (Alberta)	Mean ODR at the end of the grazing season varied between 0.24 and 0.35 between different years within the same herd. The proportion of animals with ODR > 0.5 ranged from 0.03 to 0.33 between years.
Marley et al., 2014 [20]	Serum (N.A.)	7	beef	48 (1)	UK	Mean ODR prior to winter housing of 0.7. There were no differences in FEC, dry faecal matter, or dry matter adjusted FEC, ODR, or plasma pepsinogen levels of steers grazing either chicory/ryegrass swards or ryegrass only swards.
Beck et al., 2015, [16]. Parasite Vectors 8, 434	Serum (1/140)	8	beef	1000 (26)	West Canada (Alberta)	ODR is correlated with meteorological variables (temperature, growing degree days, and humidity). ODR values of >0.5 are indicative of high exposure to gastrointestinal nematodes.
Merlin et al., 2016, [18]. Vet Parasitol 225, 61–69	Serum (1/160)	4–8	dairy	291 (12)	France	Throughout the grazing season, mean ODR values increased gradually over time, with the maximum value reached at 1.7 months after housing. A high and positive correlation was found between mean ODR and pepsinogen level. Animals with higher values of max ODR had a significantly higher risk of having lower ADWG.
Merlin et al., 2017, [21]. Prev. Vet. Med 138, 104–112	Serum (1/160)	4–17	dairy	577 (24)	France	A cut of 0.7 ODR was used to separate low and highly-exposed groups to GI nematodes. In the highly exposed groups, ADWG was negatively correlated with increasing ODR, whereas no relationship between ODR and heifer growth was seen in the lowly exposed groups.
Merlin et al., 2018, [17]. Animal 12, 1030–1040	Serum (1/160)	6–16	dairy	540 (23)	France	The same cut-off was used as the above study to define a low group (<0.7 ODR) and a highly exposed group (≥0.7 ODR). In the highly exposed group, there was a significant improvement in post-treatment ADWG in treated animals compared with untreated animals.
Constancis et al., 2022, [22]. Vet. Parasitol. 302, 109,659	Serum (1/160)	2–9	dairy	902 (44)	France	FEC, pepsinogen, and *Ostertagia* ODR were monitored at the end of the grazing season in an organic farm with a calf rearing system with nurse cows (mixed grazing of calves and adults at a ratio of 2–4 calves per nurse cow) in year 1 and at 4 occasions across the grazing season in year 2. All herd average indicators were in line with low exposure to gastrointestinal nematodes (<0.7 ODR). A decrease in ODR was observed at the second sampling occasion, likely explained by the passive transfer of antibodies from the dam.

Legend: ODR, optical density ratio (test result of *O. ostertagi* ELISA); ADWG, average daily weight gain; FEC, faecal egg count.

**Table 2 animals-13-02226-t002:** Descriptive statistics of the studied variables in first-season grazing calves on seven Flemish dairy farms and four Swedish herds.

Measure	Pepsinogen (U Tyr)	*O. ostertagia* Antibody Level (ODR)	Faecal Egg Count (EPG)	Weight (kg)
*N*	396	396	335	396
Minimum	0.1	−0.11	0	65
1st quartile	1.1	0.32	0	293
Median	1.5	0.64	0	361
Mean	1.6	0.58	20	362
3rd quartile	2.1	0.84	10	427
Maximum	8.2	1.42	1600	718
Standard deviation	0.9	0.34	147	119

**Table 3 animals-13-02226-t003:** Spearman rank correlation coefficient between *O. ostertagia* antibody levels (ODR), pepsinogen, and Faecal egg count (FEC) in three situations.

	ODR/Pepsinogen	ODR/FEC	Pepsinogen/FEC
Animal level over the entire grazing season	0.54	0.31	0.36
Animal level at the end of the grazing season	0.25	0.09	0.09
Mean level over the entire grazing season	0.74	0.64	0.75

## Data Availability

The data presented in this study are available on request from the corresponding author.

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
