# Peer review of "Review and Evaluation of Ostertagia ostertagi Antibody ELISA for Application on Serum Samples in First Season Grazing Calves"

_animals, 2023, doi:10.3390/ani13132226_

Round 1

Reviewer 1 Report

In this research, the authors wanted to review the literature in which the Svanovir® Ostertagia ostertagi-Ab ELISA has been applied to serum samples from first season grazing calves and to compare this assay with the commercial assay (detection of pepsinogen concentration).

The article is good enough with scientific interest but there are some points that have to be clarified or changed.

1. Title: the tile refers only in the evaluation of the assay but not in the review.  

Introduction

2. Aim of the study: “(ii) to compare the diagnostic value of both assays”, which “both assays” do you mean?

Methods

3. 2.2. Comparative analysis of O. ostertagi antibody and pepsinogen levels measured in sera of dairy calves

L 2: (2011) do you mean 2021??

Results

4. “3.1. Literature review

The search strings yielded 810 publications in PubMed. A title-based selection yielded 35 potentially relevant publications. A second full-text-based selection 15 publications reporting results on the use of the Svanovir® O. ostertagi-Ab ELISA on bovine sera. Of these, 10 publications applied the assay to sera coming from calves, while the remaining 5 applied it to sera from adult cattle only.”

It is better to use a flow diagram that depicts the flow of information.

5. “The 11 relevant studies are summarised in Table 1.” Why 11 studies? How did this number come out?

6. Tables and figures should be places directly after the paragraph they mentioned.

7. “3.2.3. Correlation with parasitological parameters”

“ROC analysis revealed that a threshold…” where is the figure of ROC analysis?

Author Response

Response to reviewers’ comments

We are grateful for your consideration of our manuscript for publication in Animals entitled “Evaluation of Ostertagia ostertagi Antibody ELISA for Appli-cation on Serum Samples in First Season Grazing Calves (animals-2464182)”. We appreciate the positive evaluation of our manuscript and thank you for the constructive feedback provided. We submit a revised manuscript, together with a line-by-line response to the reviewers’ comments. The reviewer comments are included in italics and our response in non-italicized text.

  1. Title: the title refers only in the evaluation of the assay but not in the review.
  • The title has been changed to ‘Review and Evaluation of Ostertagia ostertagi Antibody ELISA for Application on Serum Samples in First Season Grazing Calves’

  1. Aim of the study: “(ii) to compare the diagnostic value of both assays”, which “both assays” do you mean?
  • Deleted ‘both assays’ and add ‘ostertagi-AB ELISA and pepsinogen tests.’

  1. 2. Comparative analysis of O. ostertagi antibody and pepsinogen levels measured in sera of dairy calves L 2: (2011) do you mean 2021?
  • The date (2011) was correct. The Belgium data have never been published.

  1. “3.1. Literature review. The search strings yielded 810 publications in PubMed. A title-based selection yielded 35 potentially relevant publications. A second full-text-based selection 15 publications reporting results on the use of the Svanovir® O. ostertagi-Ab ELISA on bovine sera. Of these, 10 publications applied the assay to sera coming from calves, while the remaining 5 applied it to sera from adult cattle only.” It is better to use a flow diagram that depicts the flow of information.
  • We understand that a diagram can be useful. However, this paper is not a full systematic literature review, which means such a graph might not be necessary. We added a sentence to clarify the section and the total number of selected studies.

  1. “The 11 relevant studies are summarised in Table 1.” Why 11 studies? How did this number come out?
  • They comprise 10 studies from the PubMed search and the one extra relevant PhD dissertation from Google scholar search. More text has been added to clarify.

  1. Tables and figures should be placed directly after the paragraph they mentioned.
  • We were informed by the journal that the tables should be placed at the end of the file and figures as separate files. We believe that the journal will make the format right before publication.

  1. “3.2.3. Correlation with parasitological parameters”. “ROC analysis revealed that a threshold…” where is the figure of ROC analysis?
  • We prefer not to include the figure to keep the manuscript compact. However, the figure is provided here for the reviewer’s information.

Reviewer 2 Report

The article is well written, the data are properly presented and explained. However, there are some minor issues that require attention before the manuscript can be accepted.

 This manuscript provides an interesting contribution on the prevalence of O. ostertagi in calves using two assays in the first season of grazing.

      I believe that the manuscript can be improved if the following suggestions are considered:

-In the discussion section, how could the difference in the years of sample collection be explained?. Section 2.2.

-        - Generalized linear mixed models may be included as a supplementary file. Section 2.2.

-        -Could you enlarge Figure 2 to see the distribution of the data?

Minor editing of English language required

Author Response

Response to reviewers’ comments

We are grateful for your consideration of our manuscript for publication in Animals entitled “Evaluation of Ostertagia ostertagi Antibody ELISA for Appli-cation on Serum Samples in First Season Grazing Calves (animals-2464182)”. We appreciate the positive evaluation of our manuscript and thank you for the constructive feedback provided. We submit a revised manuscript, together with a line-by-line response to the reviewers’ comments. The reviewer comments are included in italics and our response in non-italicized text.

  1. In the discussion section, how could the difference in the years of sample collection be explained? Section 2.2.
  • This was because the Belgian study was conducted 12 years ago, and the data have never been published. The Swedish study was relative recent. We decided to analyse them together because of the high similarity in their study design.

  1. Generalized linear mixed models may be included as a supplementary file. Section 2.2.
  • We understand the reviews’ advice on including all statistical results. However, not all the results from GLMMs tables were relevant. After double-checking the text, we believe that the extracted results in Section 3.2.4 was enough to explain the key findings clearly. Full stats results can be provided upon request.

  1. Could you enlarge Figure 2 to see the distribution of the data?
  • The original figures we uploaded were in full size. It is possible that the figure shrunk due to formatting issue. We will make sure that the figure is in full size before the publication.